# ^177^Lu-PSMA Therapy in Metastatic Castration-Resistant Prostate Cancer

**DOI:** 10.3390/biomedicines9040430

**Published:** 2021-04-15

**Authors:** Yasemin Sanli, Duygu Has Simsek, Oner Sanli, Rathan M. Subramaniam, Ayse Tuba Kendi

**Affiliations:** 1Department of Nuclear Medicine, Istanbul Faculty of Medicine, Istanbul University, Istanbul 34093, Turkey; yasemin.sanli@yahoo.com (Y.S.); dr.duyguhas@hotmail.com (D.H.S.); 2Department of Urology, Istanbul Faculty of Medicine, Istanbul University, Istanbul 34093, Turkey; onersanli@hotmail.com; 3Dean’s Office, Otago Medical School, University of Otago, Dunedin 9054, New Zealand; rathan.subramaniam@otago.ac.nz; 4Department of Radiology, Mayo Clinic, Rochester, MN 55905, USA

**Keywords:** CRPC, ^177^Lu-PSMA, PSA, prostate cancer, metastasis

## Abstract

The aim of this narrative review is to evaluate the current status of ^177^Lu-PSMA (prostate specific membrane antigen) therapy for metastatic castration-resistant prostate cancer (mCRPC) in the light of the current literature. We also addressed patient preparation, therapy administration and side effect profiles. ^177^Lu-PSMA therapy efficacy was assessed by using prospective trials, meta-analyses and major retrospective trials. Predictors of efficacy were also mentioned. Although there are some different approaches regarding the use of ^177^Lu-PSMA therapy in different countries, this type of therapy is generally safe, with a low toxicity profile. From the oncological point of view, a PSA (prostate specific antigen) decline of ≥50% was seen in 10.6–69% of patients with mCRPC; whereas progression-free survival (PFS) was reported to be 3–13.7 months in different studies. Consequently, ^177^Lu-PSMA therapy is a promising treatment in patients with mCRPC, with good clinical efficacy, even in heavily pretreated patients with multiple lines of systemic therapy. Currently, there are ongoing clinical trials in the United States, including a phase III multicenter FDA registration trial.

## 1. Introduction

Prostate cancer (PCa) is the most common cancer and the second cause of death related to cancer in the United States [1]. For 2020, it was estimated that 191,930 new cases would be diagnosed; whereas a total of 33,340 cases would have died due to PCa [1]. Depending on the relationship between serum PSA (prostate specific antigen) levels and applied treatment according to the location of the disease, the natural course of PCa might be evaluated in four distinct disease states. In the initial state, disease is localized to the prostate and curative treatment options such as radical prostatectomy (RP) or radiotherapy (RT) are available. If the patient was not cured at the initial phase, a non-castrate rising PSA phase follows. The remaining states are non-castrate metastatic and metastatic castration-resistant PCa (mCRPC), respectively. It is estimated that the metastatic state of PCa leads to death in 30% of the patients within 5 years; whereas the survival of patients with mCRPC is only 14 months [1,2].

For the metastatic disease state, the backbone treatment option is hormonal. Briefly, removing the androgens (androgen deprivation therapy (ADT)) leads to the inhibition of progression of PCa for a period of 18–24 months [3]. Despite appropriate ADT, if the disease progresses, it is defined as mCRPC [4]. For this disease state, many agents given in conjunction with ADT are available with different survival benefits. For example, abiraterone, enzalutamide and apalutamide interfere with androgenic stimulation of the PCa growth by blocking androgen synthesis [5]. Meanwhile, systemic taxane-based chemotherapies, such as docetaxel for patients with high metastatic burden, or cabazitaxel after treatment with docetaxel, are also available. Immunotherapy with sipueleucel-T and bone-targeted radiopharmaceutical therapy with radium-223 (^223^Ra) are also in use, although less frequent [6]. Lastly, the use of theranostic agents such as ^177^Lu or ^225^Ac-PSMA is an emerging therapy in patients with mCRPC, available for the clinical care of patients in many countries outside North America and in a multicenter phase III clinical trial in the United States. This review aims to summarize the theranostic benefits of ^177^Lu-PSMA in mCRPC.

## 2. ^177^Lu-PSMA as a Therapeutic Agent

Prostate-specific membrane antigen (PSMA), the glutamate carboxypeptidase II, is a type II membrane glycoprotein that is physiologically expressed in several tissues. Meanwhile, it is well-known that this protein is over-expressed in PCa [7]. Currently, ^68^Ga-labelled PSMA imaging shows a high diagnostic accuracy in staging and also the detection of early biochemical recurrence with very promising performances at very low PSA levels [8,9,10]. Overexpressed in a majority of PCa cells, PSMA can allow effective use as a molecular imaging target and for targeted radioligand therapy of castration-resistant PCa [11]. A beta emitter ^177^Lu or alpha emitter ^225^Ac binding with a PSMA molecule was deemed to be an emerging radionuclide as a theranostics agent [12]. As a further management strategy, it might also be used as a theranostic radiotracer in combination with ^177^Lu and ^225^Ac to locate PCa cells and specifically destroy them. This approach, using some alpha or beta emitter agents, came into practice with favorable outcomes as a part of personalized medicine, which is highly convenient for decision-making and monitoring therapy.

## 3. Patient Preparation for ^177^Lu-PSMA Therapy

### 3.1. Patient Selection and Preparation

Briefly, patient selection, preparation and cautionary considerations are described in Table 1, based on recent EANM (European Association of Nuclear Medicine) guidelines and literature [13,14,15] (Table 1).

### 3.2. Radionuclide Preparation

The ^177^Lu-PSMA-617 and ^177^Lu-PSMA-I&T labeling protocols were described in the literature [16,17]. The standard administered activity of ^177^Lu-PSMA therapy is variable across the published literature. Administered doses can vary by 2–8 GBq for each therapy, up to 4–6 cycles at 6–12 weekly intervals [12,15,16,18,19,20,21]. A standard empiric dose of 6–7.4 MBq is mostly administered to patients due to the reported low rates of adverse events unless any risk factor is present for toxicity [15,21]. It was shown that tumor burden affects the cumulative dose of nontargeted organs, which was described earlier as the “tumor sink effect” by Beauregard et al. [22]. Depending on this idea, Violet et al. used an algorithm and reported the safety results of ^177^Lu-PSMA-617 therapy when increasing the injected dose by 1 GBq if there were >20 sites of disease, decreasing dose by 1 GBq if <10 sites, increasing dose by 0.5 GBq per factor if weight > 90 kg or GFR(glomerular filtration rate) > 90 mL/min and decreasing dose by 0.5 GBq if weight < 70 kg or GFR < 60 mL/min [20]. Finally, there is an ongoing Phase I dose escalation study with ^177^Lu-PSMA-617 using a dose fractionation regimen, presuming that the dose-escalation of ^177^Lu-PSMA-617 is safe up to 22.2 GBq per cycle with fractionated dosing, with promising early efficacy and tolerability signals (NCT03042468). In the literature, up to six cycles have been described without any serious adverse events [15]. Dosimetry can be used to calculate the cumulative dose of non-targeted organs such as kidneys, salivary/lacrimal glands or bone marrow to avoid the radiation toxicity.

## 4. Therapy-Related Issues

### 4.1. Performing Therapy

At the start of therapy, patients are advised to be well hydrated by oral intake. Oral hydration before, on the day and two days after therapy is encouraged. Injection of ^177^Lu-PSMA is administered intravenously over one to two minutes. Meanwhile, in patients with low cardiovascular risk, 1000–2000 mL 0.9% NaCl can be given after the therapy [13,17]. Corticosteroids and antiemetics can be used. Despite being controversial, ice-packs could be used for the external cooling of salivary glands, 30 min before and up to four hours after the therapy [14,23]. Due to the renal excretion of ^177^Lu-PSMA, patients should follow the radiation safety rules for contamination risk. To reduce the radiation exposure, patients are advised to void frequently or are catheterized if the condition of patient is not well enough to void [24]

### 4.2. Release of Patients

After the administration of ^177^Lu-PSMA, the patient-specific radiation dose decreases below 25 μSv/hour at 1 m, which allows treatment in outpatient settings in Australia; whereas in Europe, this requires the admission of patients to specialized shielded hospital radiation wards for 1–3 days [24,25,26]. Patients are warned to stay away from children, and especially pregnant women, for approximately 3 days after the therapy, to follow the hygiene rules for contamination risk, and are encouraged to maintain hydration, void frequently and shower daily. In the United States, ^177^Lu-PSMA therapy is performed under clinical trials and the treatments are performed usually in outpatient settings, with home release exposure below 5 mSv/hour and guidance given to patients. This guidance is similar to the ^177^Lu-DOTATATE therapy home release guidance [27].

### 4.3. Post-Therapy Imaging

A whole body scan can be performed 24–48 h post-injection to confirm tumoral uptake. SPECT/CT (single-photon emission computed tomography) can be added to whole body imaging (4, 24 and 96 h post-injection) for dosimetry analysis. In addition to PSA response, patients may be evaluated for interim therapy response with ^68^Ga-PSMA imaging after two cycles of ^177^Lu-PSMA therapy, for determining lesion response (Figure 1). The patients can be evaluated with ^68^Ga-PSMA imaging after finishing four cycles of therapy, for evaluating therapy response (Figure 2).

## 5. Dosimetry

Several studies have published radiation dosimetry results of ^177^Lu-PSMA therapy with favorable outcomes [18,20,28,29,30,31]. The most commonly used radiopharmaceutical is ^177^Lu-PSMA-617 for PSMA ligand therapy in multiple centers. ^177^Lu-PSMA-I&T, which is similar to PSMA peptide molecules with a different chemical conjugation was also defined in recent studies with comparable results [19,30]. The reported absorbed doses were 0.39–0.99 Gy/GBq for kidneys, 0.44–1.4 Gy/GBq for salivary glands, 2.8–3.8 Gy/GBq for lacrimal glands and 0.002–0.11 Gy/GBq for bone marrow, respectively. Tumor doses were approximately 3.2–13.1 Gy/GBq according to published dosimetry results [18,20,28,29,30,31]. These studies used different dosimetry methods (whole body versus SPECT/CT; MIRD (medical internal radiation dose) versus voxel-based dosimetry) or molecules (PSMA-617 versus I&T), which made the results heterogeneous. However, the toxicity associated with these organs is rarely seen in clinical practice. The usage of whole-body imaging for dosimetry could cause the overestimation of the absorbed organ doses due to the superimposition of adjacent tissues’ activities [32]. Supporting this view, previous studies which used SPECT/CT for dosimetry reported lower doses for non-targeted organs, and supported that PSMA ligand therapy is a safe treatment option with a lack of significant toxic effects [18,20]. Violet et al. published that the SUV (standardized uptake value) max and absorbed dose in salivary glands and kidneys decreased significantly if a greater disease burden existed; thus, tumor burden may be relevant in predicting salivary gland and renal toxicity [20].

The current thresholds of absorbed organ doses were defined based on external beam radiotherapy (EBRT) literature. The accepted absorbed dose is 23 Gy for kidneys, 34 Gy for lacrimal glands, 26 Gy for parotid glands and 2 Gy for bone marrow [33,34,35]. Due to the differences between EBRT and systemic radionuclide therapy, organs can tolerate higher doses in radionuclide therapy compared to EBRT, which has been demonstrated in a recent study [36]. Depending on previously mentioned dosimetry results, the absorbed doses to critical organs will be below the accepted doses, even in repeated cycles of ^177^Lu-PSMA-617 therapy [15,37]. Specific absorbed dose estimates for critical organs under ^177^Lu-PSMA-617 or ^177^Lu-PSMA-I&T therapy are summarized in Table 2.

## 6. Adverse Events

Due to the nature of the PSMA ligand therapy, unintentional radiation exposure is delivered to PSMA-expressing non-targeted tissues. However, limited side effects can be observed related to the absorbed dose. Kidneys, salivary glands, lacrimal glands and bone marrow are the most exposed organs. However, the reported adverse events were mild or transient related to ^177^Lu-PSMA therapy [12,19,21,38]. Hematotoxicity is the most common serious adverse event due to the bystander effect, which is described in 12% of patients undergoing ^177^Lu-PSMA-617 therapy with high tumor burden in bone [21]. Grade 3–4 toxicities, anemia (10%), leukopenia (3%) and thrombocytopenia (4%) were reported in the multicenter trial [21], whereas the rates were similar to the placebo group (1–14%) or ^223^Ra therapy group (3–13%) in the ALSYMPCA (ALpharadin in SYMptomatic Prostate Cancer) trial [39]. In contrast to the low hematotoxicity rates of ^177^Lu-PSMA-617 therapy, reversible hematologic toxicity including Grade 4 thrombocytopenia and Grade 4 neutropenia were reported in 46.8 (29.8% received platelet transfusions) and 25.5% of patients in a phase 2 study of ^177^Lu-PSMA-J591 therapy [40]. Xerostomia is the second most common adverse effect and was reported in 8 and 11% in two different studies [21,38]. However, mild xerostomia was found to be in 87% of patients in whom external cooling of salivary glands was not administered [16]. Grade 3–4 nephrotoxicity was not reported in current studies [14,16,21,38].

## 7. Efficacy

### 7.1. Prospective Trials

A few prospective trials are available to demonstrate the efficacy of ^177^Lu-PSMA therapy in the treatment of mCRPC with progressive disease. In a dual-center phase II trial, Tagawa et al. used ^177^Lu-PSMA-J591 therapy in a cohort of 47 patients in whom 55.3% had also received prior chemotherapy [40]. Briefly, 10.6, 36.2 and 59.6% of the patients had a PSA decline of ≥50%, ≥30% and any PSA decline, respectively, after a single dose of ^177^Lu-PSMA therapy. Meanwhile, 66.7% of the patients had a ≥50% decline in circulating tumor cell (CTC) counts. The median time for progression was 12 (8–47) weeks. When the authors made a comparison between maximum tolerated dose and survival, they found that survival was longer (21.8 vs. 11.9 months) with maximum tolerated dose (70 vs. 65 mCi/m^2^) [40].

In a phase II proof of concept trial, Hofman et al. recruited 30 patients with mCRPC with progressive disease who previously used standard therapy options such as docetaxel (87%), cabazitaxel (47%) and abiraterone, and/or enzalutamide (83%) [16]. Patients received 1 to 4 cycles of intravenous ^177^Lu-PSMA-617 at six weekly intervals. Disease progression status and adverse reactions were assessed. In addition to the primary study endpoint of PSA response rate (50% decline from the baseline), other endpoints such as progression-free survival (PFS) and overall survival (OS), as well as imaging responses and quality of life (QOL), were also evaluated. Briefly, 17 (57%) patients met the PSA response criteria and 29 (97%) of the patients had some degree of PSA response. Meanwhile, 3 months after the last injection, 40, 37, and 37% of the patients had the non-progressive disease (complete response, partial response and stable disease) according to ^68^Ga PSMA-11, FDG (fluorodeoxyglucose), and bone scanning, respectively. During follow-up, PSA progression was encountered in 27 (90%) patients and the median PFS and OS were calculated as 7.6 and 13.5 months, respectively. ^177^Lu-PSMA-617 therapy was well-tolerated with minimal adverse events, such as dry mouth, in 87% of the patients. Moreover, the pain level of the study participants (27 patients (90%)) improved in all study time points that significantly contributed to the quality of life. Even though this study had no control group for comparison, a randomized multicenter study comparing the activity and safety of ^177^Lu-PSMA-617 therapy in comparison with cabazitaxel was recently published (TheraP trial). The primary endpoint was PSA response, defined by a reduction of at least 50% from the baseline [41]. The authors showed that this PSA reduction was higher in the assigned ^177^Lu-PSMA-617 group than the cabazitaxel group (65/99 vs. 37/101 patients). Meanwhile, PFS at 12 months was shown to be better in the ^177^Lu-PSMA-617-treated group (19 vs. 3%). Additionally, Grade 3–4 adverse events occurred more frequently in the cabazitaxel group (33 vs. 53%), whereas Grade 3–4 thrombocytopenia was more common (11 vs. 0%) and Grade 3–4 neutropenia was less commonly seen in the ^177^Lu-PSMA-617-treated group (4 vs. 13%). Accordingly, despite this data providing strong evidence that ^177^Lu-PSMA-617 is more efficient than cabazitaxel, OS data was not given.

Recently, Yadav et al. reported the outcomes of ^177^Lu-PSMA-617 therapy in a prospective single arm, in a single-center study. They recruited 90 mCRPC patients with progressive disease on second-line hormonal therapy and/or docetaxel chemotherapy [38]. After the first cycle of ^177^Lu-PSMA-617 therapy and at the end of the therapy (up to seven cycles), a greater than 50% decline in PSA was shown in 32.2 and 45.5% of patients, respectively. Tumor response such as partial remission, stable disease, and progressive disease were seen in 27.5, 43.5, and 29% of patients, respectively. Additionally, improvement in the pain score was observed in 54% of patients after the first cycle of therapy, with a corresponding reduction in the analgesic score. The median OS and median PFS were 14 and 11.8 months, respectively. Similarly, a randomized phase III VISION trial was developed to compare overall survival (OS) in patients with progressive PSMA-positive mCRPC. The study group was randomized to receive either ^177^Lu-PSMA-617 in addition to the best supportive and best standard of care, or the best supportive and best standard of care alone. The estimated study completion is May 2021 (NCT03511664). The sample size was calculated as 750 patients who received at least one novel androgen axis drug, as well as having been previously treated with one to two taxane regimens.

### 7.2. Meta-Analyses

A recent meta-analysis by Kim et al. gathered available data from 10 retrospective series including 455 patients that reported favorable outcomes [42]. That meta-analysis revealed that approximately 2/3 of any PSA decline and 1/3 of more than 50% PSA decline can be expected after the first cycle of ^177^Lu-PSMA-617 therapy. Any PSA decline resulted in survival prolongation after the first cycle of therapy. Subsequently, another meta-analysis, including 10 studies, by Calopedos et al. also showed similar results, with approximately 68% of patients having had a biochemical response defined as any decline in serum PSA values and 37% of patients having had a decline >50% [43]. At subgroup analysis, the authors noted that 71% of the patients treated with ^177^Lu-PSMA-617/I&T had any PSA decline, whereas this proportion was 67% in patients treated with ^177^Lu-PSMA-J591. None of these meta-analyses presented survival data. Very recently, a review and meta-analysis was published by Von Eyben et al., with 2346 patients who had been treated with PSMA radionuclide therapy [44]. They showed that the median overall survival was 16 months, and moreover, asymptomatic patients and patients who had only lymph node metastases had longer survival than the patients who had the symptomatic and more extensive disease.

A recent systematic review compared ^177^Lu-PSMA therapy with third-line treatments such as abiraterone, enzalutamide and cabazitaxel [45]. The median serum PSA values were comparable between groups. Despite variations in terms of ^177^Lu-PSMA therapies and the number of cycles performed, overall, 44% (95% CI 31–51%) of the patients had a PSA decline of ≥50%. This was 21% (95% CI 16–27%) in patients receiving third-line treatment (overall 44 vs. 22%, *p* = 0.002). Individually, the rate of PSA decline of ≥50% was 4, 20 and 29% with abiraterone, enzalutamide and cabazitaxel, respectively. In available articles, OS was 14 months for ^177^Lu-PSMA therapy and 11 months for third line treatment alternatives (*p* = 0.32). Discontinuation due to adverse events was more common in patients receiving third-line treatment alternatives.

### 7.3. Major Retrospective Trials

Rahbar et al. retrospectively reported results from 12 therapy centers for 145 patients with serial PSA levels at baseline, and follow-up for 99 of 145 patients (68%) [21]. The response was determined as the lowest PSA level measured at follow-up and at least 8 weeks after the start of the first ^177^Lu-PSMA-617 cycle. As the major finding, 45% of patients showed a PSA decline of at least 50%. These patients were considered as biochemical responders. Accordingly, a PSA decline of any amount occurred in 60% of patients. Most responders showed a PSA decline of 50% after the first cycle. Visceral metastases and high alkaline phosphatase (ALP) levels (more than 220 U/L) were predictors of a poor outcome. Patients with three or more cycles of ^177^Lu-PSMA-617 therapy responded better. Of 47 patients, 21 (45%) had a partial response and 13 (28%) had stable disease in the follow-up period according to imaging assessments. Grade 3 and 4 hematologic toxicity occurred in 12% of the patients and no therapy-related deaths were reported.

Brauer et al. [46] and Ahmadzadehfar et al. [47] reviewed overall survival in patients receiving up to eight cycles of ^177^Lu-PSMA-617 therapy. Any PSA decline after the first cycle was a significant predictor of survival in both studies (56 vs. 29 weeks and 68 versus 33 weeks, consecutively). Meanwhile, there is some evidence that lymph node metastasis responds better to ^177^Lu-PSMA therapy than bone metastasis, which is attributed to the higher and more uniform radiation dose absorbed by lymph node metastasis [48].

Ahmadzadehfar et al. reported predictors of overall survival in 100 patients, who received a total of 347 cycles of ^177^Lu-PSMA-617 therapy [49]. Analyzing the percentage of PSA decline after the first cycle determined a PSA decline more than or equal to 14% as the best cut-off point, with a median OS of 88 versus 29 weeks. The median OS was significantly higher in patients without hepatic involvement, high levels of albumin and Hb and low levels of AST (aspartate aminotransferase). Rahbar et al. confirmed these findings in a larger two-center study in patients who had prior chemotherapy and abiraterone and/or enzalutamide [50]. The PSA decline of 20.8% was the optimal parameter in predicting improved overall survival in a multivariate analysis with a median OS of 68 vs. 44 weeks. The same investigators also showed that one-third of patients could be late responders and further therapy cycles should be performed for patients who do not respond to the first cycle [51].

Baum et al. studied 56 patients who received up to five cycles of ^177^Lu-PSMA-I&T therapy [19]. Within a follow-up period of 28 months, 12 patients died (21.4%) and survival was 78.6%. The median progression-free survival was 13.7 months. In a similar fashion, Rahbar et al. compared 28 patients receiving ^177^Lu-PSMA-I&T therapy with a historical cohort of 20 patients who received best supportive care [52]. The authors reported the number of treatment sessions resulted in a dependent decline in serum PSA values. For example, any PSA decline was observed in 59 and 75% of patients after one and two therapies, respectively. Meanwhile, the OS of 28 patients was 29.4 weeks, whereas the OS of the historical best supportive care was 19.7 weeks (HR 0.44, 955 CI 0.20–0.95, *p* = 0.031).

Recently, Barber et al. studied clinical outcomes of ^177^Lu-PSMA-617/I&T therapy in taxane chemotherapy pretreated (T-pretreated) versus taxane chemotherapy naïve (T-naïve) patients [53]. Of the 167 patients treated in this study, 83 were T-pretreated and 84 were T-naïve. T-pretreated patients had an overall poorer performance status, higher prevalence of bone metastases, higher PSA levels, lower Hb levels and higher AlkPtase levels compared to T-naïve patients. The median OS was 10.7 months for T-pretreated patients versus 27 months (*p* < 0.001) in T-naïve patients. Meanwhile, radiographic progression-free survival (rPFS) determined with ^68^Ga-PSMA PET-CT was 6 and 8.8 months (*p* < 0.001) for T-pretreated and T-naïve patients, respectively. PSA response was defined as a PSA decline of >50% from the baseline, observed in 40 and 57% (*p* = 0.054) of the patient groups, respectively. Of note, prior taxane-based chemotherapy was not an independent predictor of OS or rPFS. This favorable outcome was also supported by Emmett et al. with serial observations by using ^68^Ga-PSMA PET/CT in response to androgen blockage in patients with both castration-naïve and castration-resistant PCa [54]. The authors reported that a median reduction in SUVmax was recorded by day 9, with LHRH (luteinizing hormone releasing hormone) analogues with bicalutamide in patients with hormone-naïve disease; whereas a median of a 45% increase in intensity of PSMA was recorded by day 9 with abiraterone or enzalutamide. The authors hypothesized that this increase in ^68^Ga PSMA intensity with androgen blockage in mCRPC may lead to more effective combination treatments, such as ^177^Lu-PSMA-617/I&T therapy and abiraterone or enzalutamide.

In terms of patient-reported outcomes, ^177^Lu-PSMA-617/I&T therapy was reported to lead to pain relief in 33–70% of patients, improved quality of life in 60% and improved performance status in 74% [55]. Overall, 30–70% of patients showed a PSA decline of 50% or more with ^177^Lu-PSMA therapy. The response rate compares well with the response of chemotherapy agents (cabazitaxel and docetaxel) [24]. Efficacy results due to RECIST (response evaluation criteria in solid tumors), PERCIST (PET response criteria in solid tumors) and EORTC (European Organisation for Research and Treatment of Cancer) criteria, percentage of symptom relief and PSA decline, PFS and OS are shown in Table 3.

### 7.4. Predictors of Efficacy

This issue was recently investigated by Ferdinandus et al. by using a retrospective cohort of 40 patients with mCRPC. Before ^177^Lu-PSMA therapy, the authors obtained ^68^Ga-PSMA PET/CT and PSA decline was determined 8 weeks after ^177^Lu-PSMA-617 therapy [56]. The authors detected a significant correlation between age and better PSA response, and higher platelet count and worse PSA response. In multivariable analysis, a high platelet count and regular need for pain medication were found to be inversely associated with any PSA decline; whereas only regular need for pain medication had a negative impact on PSA decline of more than 50%. Of note, treatment response was independent of the amount of PSMA uptake. The authors explained this situation with the rapid growth of metastasis, which was beyond the PSMA uptake, and that can be explained by the different washout times of ^177^Lu-PSMA-617 in the respective metastases. On the other hand, in a similar study with a limited cohort, Emmett et al. reported that the response to ^177^Lu-PSMA-617 therapy was related to the SUV values of ^68^Ga-PSMA-PET/CT, which was performed for metastasis screening [57]. They included the patients whose lesions had no FDG uptake, and were above or equal to liver activity in PSMA uptake. Briefly, patients with a PSA response of >30% (9 of 14 (64%) patients) after a minimum of two cycles of ^177^Lu-PSMA-617 therapy had a SUVmax value of 17 ± 9 vs. 44 ± 15 (*p* < 0.007); whereas PSMA SUVmean was detected to be 6 ± 4 vs. 10 ± 4 (*p* < 0.04) for patients with or without PSA response, respectively. Meanwhile, the site of extent of metastasis did not predict treatment response. Furthermore, Michalski et al. investigated both FDG and PSMA uptakes before ^177^Lu-PSMA-617/I&T therapy. Of the study group, 18 of 54 (33%) patients had FDG+/PSMA- lesions at PET/CT scan. They showed that FDG positive and PSMA negative uptake was a negative predictor for overall survival (6.0 ± 0.5 months vs. 16 ± 2.4 months) [58]. Consequently, additional studies are needed for clarifying the role of the intensity of PSMA activity.

In the study by Barber et al., serum ALP level was found to be an independent factor of OS in T-naïve patients, which suggests the high metastatic burden of PCa [53]. This finding is line with Bräuer et al., who found that mCRPC patients with an initial ALP level < 220 U/L and a PSA decline after the first cycle had a longer OS than patients that did not meet this criteria (56 vs. 28 weeks, *p* < 0.01 and 56 vs. 29 weeks, *p* = 0.004, respectively) [46]. Kessel et al. also showed that the beneficial effects on OS are influenced by second-line chemotherapy with cabazitaxel and the presence of visceral metastases at the beginning of ^177^Lu-PSMA-617 therapy; which are both associated with tumor burden [59].

Rathke et al. evaluated the baseline and follow-up levels of serum PSA, lactate dehydrogenase (LDH) and Chromogranin A (CgA), and whether they served as response predictors for ^177^Lu-PSMA-617 therapy [60]. Of 100 patients, 35 had partial remission, 16 had stable disease, 15 had mixed response and 36 had progression of disease. The study revealed that baseline PSA had no prognostic value for response prediction, whereas LDH had the most prognostic value and elevated serum LDH levels increased the risk for progression of disease under PSMA ligand therapy. Elevated CgA demonstrated a moderate impact as a negative prognostic marker in general but was explicitly related to the presence of liver metastases.

## 8. Future Perspectives

In most of the studies evaluating the efficacy of Lu-PSMA therapy, PSA decline was used as the primary endpoint. Despite being widely accepted as a surrogate measure of tumor response to therapy, the reliability of this endpoint is controversial [61]. For example, in a TAX-327 study, despite demonstrating a similar PSA response, a weekly docetaxel regime did not show a survival benefit in comparison with 3-weekly schedules [62]. On the contrary, in the study by Barber et al., a significant survival benefit was obtained in T-pretreated vs. T-naïve patients, and the PSA response due to ^177^Lu-PSMA-617/I&T therapy was not significant [53]. It is also worth noting that, in almost all studies evaluating the efficacy of ^177^Lu-PSMA therapy, the association between PSA decline and survival benefit was not available. For this reason, we need well-conducted prospective studies including objective endpoints to fully understand the survival benefit of ^177^Lu-PSMA therapy.

Currently, ^177^Lu-PSMA therapy is used as the last resort in the treatment of mCRPC. In almost all studies, patients had passed a long way, including different alternatives such as; docetaxel, cabazitaxel, abiraterone, enzalutamide and ^223^Ra. However, it is currently not well-known whether treatment response could be better if ^177^Lu-PSMA therapy was used in the first-line or earlier settings. Comparative studies are urgently needed.

Despite the low data yield for ^177^Lu-PSMA therapy, we believe that the future is bright about the use of theranostics in mCRPC, because theranostics for PCa will probably be a good partner for combination therapies. Combination strategies with the idea of using drugs having different efficacy mechanisms are generally deemed to be more effective compared with individual drug administrations, and have been used in clinical practice for many types of cancers, including PCa. For example, CHAARTED (ChemoHormonal therapy versus androgen ablation randomized trial for extensive disease in prostate cancer) and STAMPEDE (Systemic Therapy in Advancing or Metastatic Prostate cancer: Evaluation of Drug Efficacy) studies resulted in promising results of combining therapies in metastatic PCa [63,64]. In a similar fashion, Murga et al. also reported that the inhibition of androgen receptor pathways can increase PSMA expression, which leads to new combination regimes [65]. Supporting this idea, Basu et al. mentioned that results were gratifying when a second-generation antiandrogen therapy, in particular abiraterone, was combined with ^177^Lu-PSMA. Authors also mentioned adding a low-dose corticosteroid to obviate any possible symptomatic flare response and to prevent the mineralocorticoid excess of abiraterone [66]. Finally, idronoxil, an inhibitor of external NADH oxidase type 2, is used for radiosensitization in ^177^Lu-PSMA therapy. Crumbaker et al. published a phase 1 clinical trial of ^177^Lu-PSMA-617 plus idronoxil, with no significant toxicities or safety concerns as of yet [67]. Accordingly, with its relatively low toxicity profile and adequate efficacy in recent studies, ^177^Lu-PSMA has potential to be used as a part of combination therapies.

## 9. Conclusions

^177^Lu-PSMA therapy is a promising treatment alternative in patients with mCRPC, with good clinical efficacy, even in heavily pretreated patients with multiple lines of systemic therapy. Additionally, the available data regarding ^177^Lu-PSMA therapy revealed that this type of therapy is safe, with a low toxicity profile. There is also some preliminary evidence that ^177^Lu-PSMA therapy is more effective, if used prior to other systemic therapies, earlier during the disease course. Consequently, this treatment alternative may shift its place from the last treatment step of mCRPC to one of the initial therapy steps for PCa, probably combined with other systemic treatment options in the future.

## Figures and Tables

**Figure 1 biomedicines-09-00430-f001:**
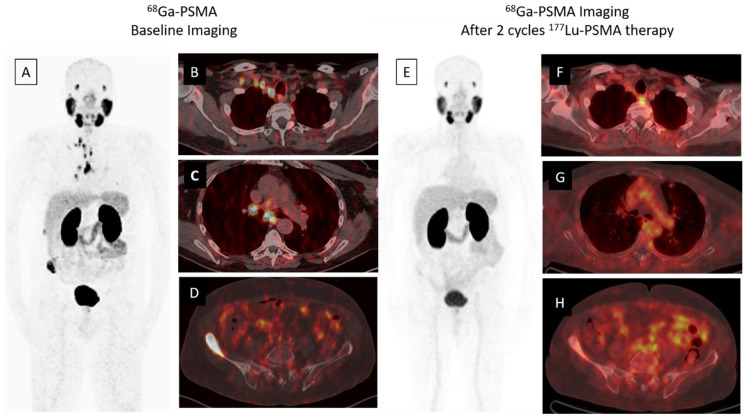
A 68-year-old male was diagnosed, with a Gleason score of 3 + 4 = 7, with prostate adenocarcinoma in 2001, who then underwent external beam radiation therapy with medical castration therapy. He had biochemical recurrence starting in November 2014. He started medical castration therapy and finally his PSA (prostate specific antigen) level slowly progressed to 7.5 ng/mL in December 2018. ^68^Ga-PSMA-617 PET/CT maximum intensity projection (MIP) image (**A**) demonstrates widespread abnormal activity in the supraclavicular (**B**) and mediastinal (**C**) lymph nodes, pelvic lymph nodes, prostate gland and bones (**D**). The patient was evaluated after two cycles of ^177^Lu-PSMA-617 therapy for response to radioligand therapy. A significant response in skeletal and lymph node metastases was detected in ^68^Ga PSMA-617 PET-CT MIP (**E**) and axial fused images (**F**–**H**) and serum PSA value decreased by approximately 80% (PSA value decreased from 7.5 to 1.53 ng/mL).

**Figure 2 biomedicines-09-00430-f002:**
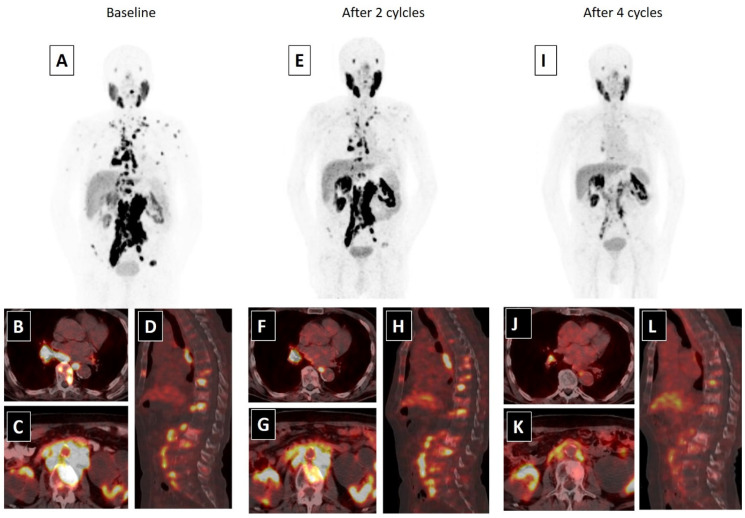
An 80-year-old male with mCRPC, having a Gleason score of 4 + 5 = 9 and a serum PSA value of 293.5 ng/mL, was not responsive to the systemic therapies, such as docetaxel, abiraterone and cabazitaxel. He was also suffering from pain (5/10) and fatigue (ECOG score 2). ^68^Ga-PSMA-617 PET-CT, ^99m^Tc-MAG3 renography and routine laboratory tests were performed to evaluate the patient for ^177^Lu-PSMA-617 therapy. In ^68^Ga-PSMA-617 PET-CT (**A**), an intense PSMA uptake was detected in supra-infra diaphragmatic metastatic lymph nodes and sclerotic bone metastases (axial fused: (**B**,**C**); sagittal fusion: (**D**)). Laboratory tests were within normal limits except for a high-normal serum creatinine level (1.4 mg/dL). ^177^Lu-PSMA-617 therapy was planned with a dosimetric approach and he received two cycles of ^177^Lu-PSMA-617 therapy (cumulative dose: 12.5 Gbq) without any significant adverse effect. A partial response in skeletal and lymph node metastases were detected in ^68^Ga-PSMA-617 PET-CT and his serum PSA value decreased by 76.2% (PSA: 69.5 ng/mL) after two cycles of therapy (MIP image: (**E**); axial fused: (**F**,**G**); sagittal fusion: (**H**)). His pain significantly resolved (3/10) and his quality of life improved (ECOG: 1). Accordingly, the patient completed four cycles of ^177^Lu-PSMA-617 therapy (cumulative dose: 25 Gbq), without any serious side effects. After the end of ^177^Lu-PSMA-617 therapy, an approximately 95% PSA decline (PSA: 13.5 ng/mL) was detected and a marked response was observed in post-therapy ^68^Ga-PSMA-617 PET-CT (MIP image: (**I**); axial fused: (**J**,**K**); sagittal fusion: (**L**)). The patients’ pain was relieved for 15 months and he was alive at 18 months of the ^177^Lu-PSMA-617 therapy.

**Table 1 biomedicines-09-00430-t001:** Summary of patient selection, preparation and cautionary considerations.

Patient selection	Patients with mCRPC who are ineligible or finalized the approved alternative options and with adequate uptake of PSMA ligands on the basis of a pre-therapy imaging study can be considered for treatment [13].
	Uptake of tumors > liver uptake (at least 1.5 times the SUV_mean_) [16]. Liver metastases negative on PSMA-ligand PET should be ruled out, even if the remainder of the disease demonstrates intense PSMA expression [13].
	Life expectancy > 6 months ECOG performance status > 2 Unless the main objective is alleviating suffering from disease-related symptoms [13].
Patient preparation and cautionary considerations	Complete blood tests need to be performed within the two weeks before the ^177^Lu-PSMA therapy
	White blood cells > 2500/L Platelet > 75,000/L Hemoglobulin > 8 mg/dL If blood cell counts were below the suggested thresholds, blood cell transfusion can be considered to avoid adverse effects [13].
	Myelosuppressive therapies should be discontinued for protecting bone marrow reserves [14].
	Patients with obstructive urinary disorders which might be evaluated with 99m Tc-MAG3 or 99m Tc-DTPA scintigraphy should be resolved before the therapy to reduce the radiation exposure to the kidneys [15].
	Creatinine level should <2× upper limit of normal GFR > 30 mL/min [13].
	Liver transaminase levels should be <5× upper limit of normal [13].

mCRPC, Metastatic castration-resistant prostate cancer; PSMA, Prostate specific membrane antigen; SUV, Standardized uptake value; PET, Positron emission tomography; ECOG, Eastern Cooperative Oncology group; GFR, Glomerular filtration rate.

**Table 2 biomedicines-09-00430-t002:** Specific absorbed dose estimates for critical organs under ^177^Lu-PSMA-617 or I&T therapy.

Reference Study	Patient Number	Molecule	Imaging Method	Dose Convolution	Kidneys	Salivary Glands	Lacrimal Glands	Bone Marrow	Liver	Spleen	Tumors
Delker [28]	5	617	Whole body planar + SPECT-CT	MIRD *	0.6 Gy/GBq	1.4 Gy/GBq	-	0.012 Gy/GBq	0.1 Gy/GBq	0.1 Gy/GBq	13.1 Gy/GBq
Hohberg [29]	9	617	Whole body planar	MIRD	0.53 Gy/GBq	0.72 Gy/GBq	2.82 Gy/GBq	-	-	-	-
Okamato [30]	18	I&T	Whole body planar	MIRD	0.72 Gy/Gbq	0.55–0.64 Gy/GBq	3.8 Gy/GBq	-	0.12 Gy/GBq	-	3.2 Gy/GBq
Fendler [18]	15	617	SPECT	MIRD	0.5–0.6 Gy/GBq	1.0 Gy/GBq	-	0.002 Gy/Gbq	0.1 Gy/Gbq	0.1 Gy/Gbq	6.1 Gy/GBq
Yadav [31]	26	617	Whole body planar	MIRD	0.99 Gy/GBq	1.24 Gy/GBq	-	0.048 Gy/GBq	0.36 Gy/GBq	-	10.94 Gy/GBq
Violet [20]	30	617	SPECT-CT	Voxel based and MIRD	0.39 Gy/GBq	0.44–0.58 Gy/GBq	-	0.11 Gy/GBq	0.1 Gy/GBq	0.06 Gy/GBq	11.5 Gy/GBq

* MIRD: Model-based estimate of absorbed dose; SPECT, Single-photon emission computed tomography; -: not available.

**Table 3 biomedicines-09-00430-t003:** Efficacy of ^177^Lu-PSMA therapy in literature.

Reference Study	Patient Number	Response % (RECIST)	Response % (Symptom)	PSA Decline > 50%	PFS	OS
Hofman [16]	30	CR: 40% PR: 37% SD: 37%	37%	57%	7.6 months	13.5 months
Baum [19]	56	CR: 20% PR: 52% SD: 28%	33%	59%	13.7 months	-
Rahbar [21]	145	PR: 45% * SD: 28% *	-	45%	-	-
Yadav [38]	90	PR: 23% SD: 54% PD: 23%	54%	45.5%	11.8 months	14 months
Tagawa [40]	47			10.6%	3 months	-
Kim [42]	455	-	-	34.45%	-	-
Calopedos [43]	369	-	-	37%	-	-
Von Eyben [45]	669	-	-	43%	-	-
Bräuer [46]	59	-	-	53%	4.5 months	8 months
Ahmadzadehfar [47]	52	-	-	44.2%	-	15 months
Kulkarni [48]	119	-	-	57.5%	10.7 months	-
Ahmadzadehfar [49]	100	-	-	69%	-	15 months
Rahbar [50]	104	-	-	49%	-	14 months
Rahbar [51]	71	-	-	56%	-	-
Rahbar [52]	28	-	-	50%	-	7.3 months
Barber [53]	167	-	-	40% ** 57% ***	6 months ** 8.8 months ***	10.7 months ** 27.1 months ***
Ferdinandius [56]	40	-	-	32.5%	-	-
Overall	28–669	CR: 20–40% PR: 23–52% SD: 28–54% PD: 23%	33–54%	10.6–69%	3–13.7 months	7.3–27.1 months

* Evaluated in 47 patients; ** Taxane-pretreated patients; *** Taxane-naïve patients; CR: Complete response; PR: Partial response; SD: Stable disease; PD: Progressive disease; PFS: Progression free survival; OS: Overall survival; RECIST: Response evaluation criteria in solid tumors; -, not applicable.

## Data Availability

Not applicable.

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
