# Peer review of "177Lu-PSMA Therapy in Metastatic Castration-Resistant Prostate Cancer"

_biomedicines, 2021, doi:10.3390/biomedicines9040430_

Round 1

Reviewer 1 Report

Thank you for submitting a good manuscript.
It is well organized and give readers a chance to enjoy a good review about an option for the treatment of patients with mCRPC.

Author Response

We thank the reviewer for positive feedback and helpful comments. We corrected the typing errors as the best we could do.

Reviewer 2 Report

In the article under review, the authors present a critical retrospective view of clinical studies in which 177LuPSMA has been used in the treatment of mCRPC. The authors clearly and concisely describe the results achieved during the last few clinical studies. The topic is highly topical and important especially for the field of nuclear medicine.

There are a few minor bugs in the text that need to be corrected:

line 124 and 127, should be 68Ga instead of 68Ga

line 132, should be 99mTc instead of 99mTc

line 240, should be 177Lu-PSMA instead of 177Lu-PSMA

line 393, should be 223Ra instead of Ra223

I have also few questions and comments:

  1. The often mentioned prophylaxis of salivary glands was botox injections. To what extent are they used in clinical studies?
  2. In therapy, it is very difficult to set the exact required dose. Very often an empirical estimate of the 177Lu-PSMA dose is made. It is even more difficult to make an accurate dose estimate for alpha sources.
  3. Table 2 ought to be near the second paragraph around line 172.
  4. line 111, 5 mrem/hour should be also posted in mSv/hour
  5. Structural formulas od PSMA DKZ/I&T, J591 and 617 and 11 ought to be placed in the text. It might help in better orientation in text.
  6. Could you also commnet the relation between PSA level and SUV/SUVmax or tumor to liver ratio TRL in evaluation of treatment progress? 

I recommend accepting the article for printing after minor revisions.

Author Response

We would like to express our sincere gratitude to you for the opportunity to revise our manuscript. The paper has been revised to address the reviewer comments, which are appended at the end of this letter. The revision has been developed in consultation with all coauthors, and each author has given approval to the final form of this revision. The agreement form signed by each author remains valid.We hope the revised manuscript is accepted for publication in Journal.

Sincerely yours,

A.Tuba Kendi,
